# Expression, Purification and Characterization of Chondroitinase AC II from Marine Bacterium *Arthrobacter* sp. CS01

**DOI:** 10.3390/md17030185

**Published:** 2019-03-20

**Authors:** Yangtao Fang, Suxiao Yang, Xiaodan Fu, Wancui Xie, Li Li, Zhemin Liu, Haijin Mou, Changliang Zhu

**Affiliations:** 1Laboratory of Applied Microbiology, College of Food Science and Engineering, Ocean University of China, Qingdao 266003, China; FYoungT@163.com (Y.F.); yangsuxiao66@163.com (S.Y.); luna_9303@163.com (X.F.); llcs0229@163.com (L.L.); abc76216@126.com (Z.L.); 2Shandong Provincial Key Laboratory of Biochemical Engineering, College of Marine Science and Biological Engineering, Qingdao University of Science and Technology, Qingdao 266042, China; xiewancui@163.com

**Keywords:** chondroitinase, eukaryotic expression, enzymatic properties, oligosaccharides

## Abstract

Chondroitinase (ChSase), a type of glycosaminoglycan (GAG) lyase, can degrade chondroitin sulfate (CS) to unsaturate oligosaccharides, with various functional activities. In this study, ChSase AC II from a newly isolated marine bacterium *Arthrobacter* sp. CS01 was cloned, expressed in *Pichia pastoris* X33, purified, and characterized. ChSase AC II, with a molecular weight of approximately 100 kDa and a specific activity of 18.7 U/mg, showed the highest activity at 37 °C and pH 6.5 and maintained stability at a broad range of pH (5–7.5) and temperature (below 35 °C). The enzyme activity was increased in the presence of Mn^2+^ and was strongly inhibited by Hg^2+^. Moreover, the kinetic parameters of ChSase AC II against CS-A, CS-C, and HA were determined. TLC and ESI-MS analysis of the degradation products indicated that ChSase AC II displayed an exolytic action mode and completely hydrolyzed three substrates into oligosaccharides with low degrees of polymerization (DPs). All these features make ChSase AC II a promising candidate for the full use of GAG to produce oligosaccharides.

## 1. Introduction

Chondroitin sulfate (CS), a glycosaminoglycan (GAG), is a linear polysaccharide with repeated disaccharide units of β-d-glucuronic acid (GlcA) and *N*-acetyl-β-d-galactosamine (GalNAc), which are linked through a β-1,4-glycosidic bond [1,2]. Both GlcA and GalNAc residues can carry different numbers of sulfate groups in different positions. CS is classified into different types, namely CS-A, [GlcAβ1-3GalNAc(4S)]; CS-C, [GlcAβ1-3GalNAc(6S)]; CS-D, [GlcA(2S)β1-3GalNAc(6S)]; and CS-E, [GlcAβ1-3GalNAc(4S,6S)] [3]. In addition, some GlcA residues are epimerized into Iduronic acid (IdoA), and those containing repeating disaccharide units [IdoAα1-3GalNAc(4S)] are designated as dermatan sulfate (DS, CS-B) [4]. CS is mainly used for the treatment of osteoarthritis and inflammation. In Europe, CS is marketed as a slow-acting anti-osteoarthritis drug and is widely used to relieve pain and other symptoms in arthritic disease [5]. Recent studies have documented the potential use of CS as an anti-viral and anti-infective agent, as well as for tissue regeneration and engineering [6]. However, due to the large molecular weight and charge densities of CS, concerns have been raised about their possible intestinal malabsorption, which may compromise their therapeutic utility [7]. Recent studies have shown that reducing the molecular weight of CS could improve the intestinal absorption of polysaccharides [8,9].

Low molecular weight chondroitin sulfate (LMWCS), produced by the depolymerization of CS, has been shown to promote the absorption and ultimately increase the uptake of iron through LMWCS-mediated inorganic iron complexes in Caco-2 cells [10]. In comparison to CS, LMWCS conjugated with α-linolenic acid significantly enhanced the oral absorption and elimination half-life in rats after intra-gastric administration [11]. The extracellular accumulation of amyloid β proteins (Aβs) in neuritic plaque is one of the hallmarks of Alzheimer’s disease (AD) [12]. LMWCS is capable of passing through the blood–brain barrier [13] and exerting neuro-protective activities against the toxic effects induced by Aβs both in vitro and in vivo. These compounds might be useful as prophylactic and therapeutic compounds for the treatment of Alzheimer’s disease [14].

Enzymatic digestion has been an effective method for the preparation of bioactive CS oligosaccharide. In contrast to vertebrates, chondroitinase (ChSase) from microorganisms utilizes a β-eliminative mechanism to breakdown CS, splitting β-1,4-glycosidic bonds between β-d-GlcA and *N*-acetyl-β-d-GalNAc with the concomitant formation of an unsaturated C4–C5 bond within the GlcA that is located at the non-reducing end [15]. Chondroitinase AC (ChSase AC, EC 4.2.2.5) can use CS-A, CS-C, and hyaluronic acid (HA) yielding oligosaccharide products, mainly disaccharide, tetrasaccharide and hexasaccharide [16]. ChSase AC can be divided into two types—ChSase AC I and ChSase AC II. Presently, ChSase AC I is found in *Flavobacterium heparinum* and shows a random endolytic mode of action by cleaving polysaccharide substrates. On the other hand, the second enzyme, ChSase AC II shows an exolytic mode of action by cleaving disaccharides from the non-reducing end of the polysaccharide chains one at a time [17]. Since the Seikagaku Corporation in Japan, the sole supplier of ChSase AC II, ceased production in 2011, subsequent in-depth research and analysis of the enzyme by the glycobiological research community has became very limited [18].

In this study, a new ChSase AC II, was expressed and purified from a newly isolated marine bacterium *Arthrobacter sp*. CS01. The characterization and analysis of degradation products were also performed. These products were suggested to have the potential for medical and industrial applications.

## 2. Results and Discussion

### 2.1. Isolation and Identification of Strain CS01 

The different strains were isolated from the gut of the sea cucumber. After detecting enzyme activity, the strain CS01 showed the highest activity and was selected for further research. The 16S rDNA sequence of the strain CS01 was sequenced and submitted to GeneBank (accession no. MK459466.1). Analysis of the alignment of 16S rDNA gene sequences revealed that the identity of strain CS01 exhibited a 99% similarity to the *Arthrobacter* sp. LHYM225. According to the phylogenetic position of 16S rDNA (Figure 1), CS01 was assigned to the genus *Arthrobacter* sp. and named *Arthrobacter* sp. CS01.

### 2.2. Cloning and Expression of Recombinant ChSase AC II

By analyzing the genomic DNA sequence of *Arthrobacter* sp. CS01, we found an open reading frame (ORF, orf00115_1) consisting of 2370 bp called ChSase AC II. The recombinant plasmids, pPIC9K-ChSase AC II and pPICZαA-ChSase AC II, were successfully constructed and confirmed by DNA sequencing and transformed into various *P. pastoris* for protein expression. The results showed that the enzyme activity of the recombinant ChSase AC II was highest in the *P. pastoris* X33 compared to the other strains (Figure 2). This might be because *P. pastoris* X33 was a suitable strain that could effectively regulate the expression of ChSase AC II, and the recombinant ChSase AC II could be correctly folded.

### 2.3. Purification of Recombinant ChSase AC II

ChSase AC II was purified using a HiTrap^®^ Q Fast Flow (HiTrap QFF) column, with specific activity of 18.7 U/mg and 53.4% recovery (Table 1). Sodium dodecyl sulfate polyacrylamide gel electrophoresis (SDS-PAGE) showed that the enzymes were purified as a single protein band (Figure 3). The molecular weight of the purified ChSase AC II was estimated to be 100 kDa, which was much higher than the theoretical molecular weight (84 kDa). This indicated that when ChSase AC II was expressed in *P. pastoris* X33, it probably experienced post-translational modifications to form the mature protein products [19]. 

### 2.4. Substrate Specificity and Kinetic Parameters of Recombinant ChSase AC II

Three kinds of substrates were used to evaluate the substrate specificity of ChSase AC II. As shown in Table 2, the relative activity of ChSase AC II towards CS-C was found to be 100%, while the relative activity towards CS-A and HA was 115.64% and 294.93%, respectively. The V_max_ of ChSase AC II against CS-A, CS-C and HA was 186, 168 and 472 µM/min, respectively. The K_m_ of ChSase AC II against CS-A, CS-C and HA was 2.44, 1.68 and 0.165 µM, respectively. Both the relative activity and V_max_ showed that ChSase AC II was more active towards HA compared to CS-A and CS-C. It also had a much lower K_m_ value towards CS-A and CS-C, which indicated that it possessed a higher affinity towards the HA substrates. 

### 2.5. Biochemical Properties of Recombinant ChSase AC II

The effects of temperature, pH, metal ions and surfactants on the enzyme activity of ChSase AC II were further determined. The ChSase AC II enzyme activity gradually improved with a rise in temperature from 25 °C to 37 °C and was optimum at 37 °C (Figure 4A). This was similar to the enzyme activities of ChSase from *F. heparinum* (40 °C) [20], *Bacteroides thetaiotaomicron* (37 °C) [21], *Bacteroides stercoris* (37 °C) [22], and *Sphingomonas paucimobilis* (40 °C) [4]. The enzyme activity was stable below 35 °C and possessed above 80% activity after incubation at 37 °C for 180 min (Figure 4B). ChSase AC II displayed the maximum activity at pH 6.5 (Figure 4C) and retained more than 80% activity at a broad pH range of pH 5–7.5 after incubation for 6 h (Figure 4D). The optimal pH was similar to that of ChSase from *B. thetaiotaomicron* (7.1) [23], *F. heparinum* (6.8) [20], *B. thetaiotaomicron* (7.6) [21], and *S. paucimobilis* (6.5) [4].

To determine the effects of metal ions and surfactants on ChSase AC II, the ions were chelated with the enzyme at 4 °C for 6 h. The relative activities of ChSase AC II were slightly inhibited by Zn^2+^, but strongly inhibited by Hg^2+^ which reduced it to 43.2%. However, Mn^2+^ markedly increased the activity to 115.78% (Figure 4E). A certain affinity of the metal ions for the SH, CO and NH groups of amino acids was found. A change in the tertiary structure of the enzyme might lead to metal ion inhibition [24].

### 2.6. Analysis of Reaction Mode and Products 

As shown in Figure 5, the degradation products of the three kinds of substrates by ChSase AC II were analyzed by thin layer chromatography (TLC) plate. With progress in the hydrolysis, the distribution of the degradation products was found to be similar. There was only one main oligosaccharide product that appeared for CS-A, CS-C, and HA, respectively. The results indicated that ChSase AC II might degrade the substrates in an exolytic manner. The reaction products were further identified by a negative electrospray ionisation mass spectrometry (ESI-MS). As shown in Figure 6, ChSase AC II completely hydrolyzed the three substrates into oligosaccharides that had low degrees of polymerization. A mass spectrogram displayed the main peaks with a *m/z* (*z* = 1) of 458.1, 458.2 and 779.4, which corresponded to the unsaturated disaccharide of CS-A (△Di-4S − H)^—^, CS-C (△Di-6S − H)^—^, and unsaturated tetrasaccharide of HA (o-HA4 − 2H + Na)^—^, respectively. To further confirm the changes in product formation and substrate consumption during degradation, the hydrolyzed samples were analyzed by high performance liquid chromatography (HPLC). As shown in Figure 7, during CS-C degradation by ChSase AC II, only one peak with a retention time of 18.990 min increased with the reaction time, and the substrate peak (retention time of 15.709 min) decreased. 

## 3. Materials and Methods 

### 3.1. Strains, Plasmids, Reagents and Medium

The CS-degrading bacterial strains used in this study were isolated from the gut of sea cucumbers. *Escherichia coli* DH5α was used in all the cloning experiments. The plasmid pPIC9K, pPICZαA, wild-type *P. pastoris* strain GS115, KM71, and X33 were used for the eukaryotic expression of ChSase AC II. The different media used for strains cultivation are listed in Table 3. CS-A (MW: 45 kDa), CS-C (MW: 63 kDa), and HA (MW: 1200 kDa) were purchased from Bomei Biotechnology Co. Ltd. (Hefei, China). All the chemicals were of reagent grade. 

### 3.2. Isolation and Identification of Strain CS01 

The samples from the Weihai Sea Cucumber Farm (Shandong, China) were immersed, homogenated, and diluted with sterilized seawater and spread on selective agar plates. The plates were incubated for 3 days at 28 °C to form detectable colonies. At least 100 strains were inoculated into an agar-free selective medium, and the ChSase activity in the culture supernatant was assayed. Among the isolates, the most active strain CS01 was selected for further studies. Genomic DNA from strain CS01 was extracted using a total DNA extraction kit (Sangon, Shanghai, China), and then sequenced in Lingen Biotechnology Co., Ltd. (Shanghai, China). The 16S rDNA sequence was blasted and aligned with closely related sequences retrieved from GenBank using the BLASTn (National Center of Biotechnology Information, Bethesda, MD, USA) and CLUSTAL X (Conway Institute UCD Dublin, Dublin, Ireland) program, respectively, and the phylogenetic tree was constructed with the MEGA 4.0 software (Biodesign Institute, Arizona State University, Tempe, AZ, USA).

### 3.3. Cloning and Expression of Recombinant ChSase AC II

To obtain the eukaryotic expression vector, the ChSase AC II gene was amplified by PCR using the following primers—upstream primer: 5’-CCGGAATTCATGACGCACGAAGTATCCCGACG-3’ (the underlined sequence is the position of an *Ecor* I site) and downstream primer: 5’-ATTTGCGGCCGCCTAGCGGTGCAGCGTGACCTC (the underlined sequence is the position of a *Not* I site). The purified ChSase AC II fragment was digested with *Ecor* I/*Not* I and ligated into the downstream primer of alcohol oxidase 1 promoter (AOX1) in the vector pPIC9K and pPICZαA. These were termed pPIC9K-ChSase AC II and pPICZαA-ChSase AC II, respectively. After digestion by the restriction enzyme *Pme* I, the linearized form of the plasmid pPIC9K-ChSase AC II was transformed into *P. pastoris* GS115 and KM71 by electroporation at 1.5 kV in a 0.2 cm gap electroporation cuvette, while the plasmid pPICZαA-ChSase AC II was transformed into *P. pastoris* X33. The recombinant plasmid pPIC9K-ChSase AC II was selected on MD plates containing 2 mg/mL Geneticin selective antibiotic (G418), and the recombinant plasmid pPICZαA-ChSase AC II was selected on YPDS plates containing 100 μg/mL zeocin. To express the ChSase AC II in *P. pastoris*, recombinants were grown in 50 mL YPD medium at 28 °C and 200 rpm. When OD_600_ reached approximately 4.0, the cells were collected by centrifugation at 4 °C and 10,000 rpm for 10 min, re-suspended in 200 mL BMGY medium and then incubated at 28 °C and 200 rpm for 72 h. Methanol was added to a final concentration of 0.5% at every 24 h to maintain the induction of ChSase AC II.

### 3.4. Purification of Recombinant ChSase AC II

The cells were centrifuged at 4 °C and 10,000 rpm for 10 min, and the fermentation supernatant was filtered using 0.22 μm filters. ChSase AC II was purified using a HiTrap QFF column (GE Healthcare, USA). The purity of the fractions was assessed using SDS-PAGE, and the protein concentration was determined using Coomassie Brilliant Blue staining. 

### 3.5. Assay of Enzyme Activity

ChSase AC II enzyme activity was measured by increasing the absorbance to 232 nm until double bonds were formed in the reaction. CS degradation was assessed using 20 mM Tris-HCl, pH 7.5, at 37 °C. A millimolar absorption coefficient of 5.1 L/(mol·cm) was used in the calculations. One international unit was defined as the amount of protein needed to form 1 µmol of 4,5-unsaturated uronic acid/min [25].

### 3.6. Kinetic Parameters of the Recombinant ChSase AC II

The kinetic parameters, V_max_ and K_m_ of ChSase AC II were determined using a 0.03125–2 mg/mL substrate of CS-A, CS-C, and HA. The assay of enzyme activity was defined as described previously. The K_m_ and V_max_ values were calculated by using the linear regression plots of Lineweaver and Burk.

### 3.7. Biochemical Properties of ChSase AC II

The optimal temperatures of ChSase AC II were found by measuring its activity to be from 25 °C to 60 °C. The optimal pH was determined to be from 4.5 to 7.0 in the 20 mM Na_2_HPO_4_-citric acid buffer and from 7.0 to 9.0 in the 20 mM Tris-HCl buffer. To determine the thermostability of ChSase AC II, the residual activity was measured after incubating it at various temperatures (25 °C–50 °C) for 180 min. Meanwhile, the pH stability depended on the residual activity after the enzyme was incubated in buffers with different pH (4.5–9.0) for 6 h. The effect of metal ions and surfactants on the enzyme activity were analysed by incubating the purified enzyme at 4 °C for 6 h in the presence of the following reagents—10 mM KCl, NaCl, FeCl_2_, FeCl_3_, MnCl_2_, ZnCl_2_, CoCl_2_, BaCl_2_, HgCl_2_, CdCl_2_, MgCl_2_, CaCl_2_ and CuCl_2_, 0.5% (v/v) SDS, Tween-80 and EDTA.

### 3.8. Analysis of Reaction Mode and Products 

To investigate the mode of action, the reaction mixtures (1 mL) containing 1 μg purified enzyme and 2 mg substrates (CS-A, CS-C, HA) were incubated at 37 °C for 0, 5, 15, 30 min, 1, 2, 3, 4, 5 h. The degradation products were first analyzed by TLC with the solvent system (N-butanol/acetic acid/ethyl acetate/water 2:1:1:1) and visualized by heating the TLC plate at 120 °C for 10 min after spraying with 10% (v/v) sulfuric acid in ethanol. To further determine the composition and degree of polymerization (DP) of the products, the degradation products were analyzed by negative-ion ESI-MS (Agilent 1290 Infinity II-6460, Frag = 175.0V, *m/z* 100–2000 amu). To determine the changes in product formation and substrate consumption during degradation, samples taken at different reaction times (0 min, 15 min, 1 h, 3 h, 6 h) were analyzed by the HPLC using PL aquagel-OH 30 column (Agilent, Santa Clara, CA, USA) under the following conditions—the mobile phase used a buffer composed of NaNO_3_ (0.2 M) and NaH_2_PO_4_ (0.01 M), the column temperature was 25 °C, the flow velocity was 0.5 mL/min, and a refractive index detector (RID) was used.

## 4. Conclusions

The ChSase AC II from a newly isolated marine strain (*Arthrobacter* sp. CS01) was cloned, expressed, purified, and characterized. The enzyme showed maximal activity at 37 °C and pH 6.5. It was stable at a broad pH range (5–7.5) and retained above 80% activity after incubation at 37 °C for 180 min. It had broad substrate specificity towards CS-A, CS-C and HA, suggesting the potential for it to be used as a candidate for industrial applications to produce oligosaccharides. In addition, further works should be focused on improving the enzyme activity and determining the catalytic direction of ChSase AC II.

## Figures and Tables

**Figure 1 marinedrugs-17-00185-f001:**
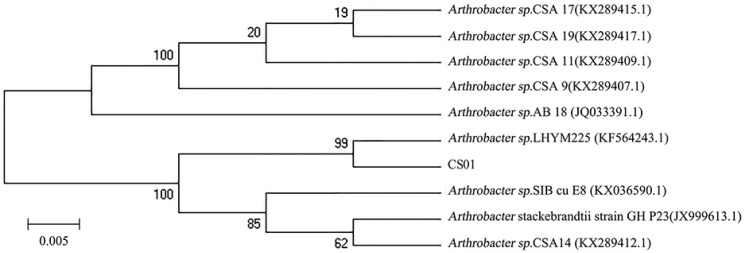
The phylogenetic tree of strain CS01 and related bacteria based on the maximum parsimony analysis of 16S rDNA sequences.

**Figure 2 marinedrugs-17-00185-f002:**
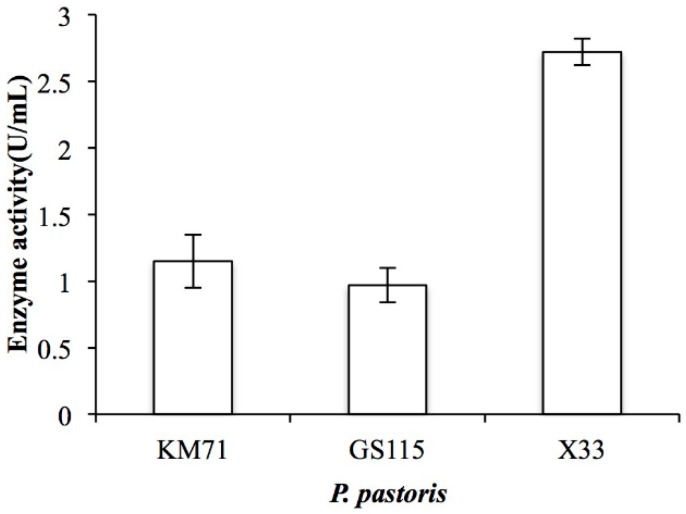
Enzyme activity of recombinant ChSase AC II in *P. pastoris* KM71, *P. pastoris* GS115 and *P. pastoris* X33.

**Figure 3 marinedrugs-17-00185-f003:**
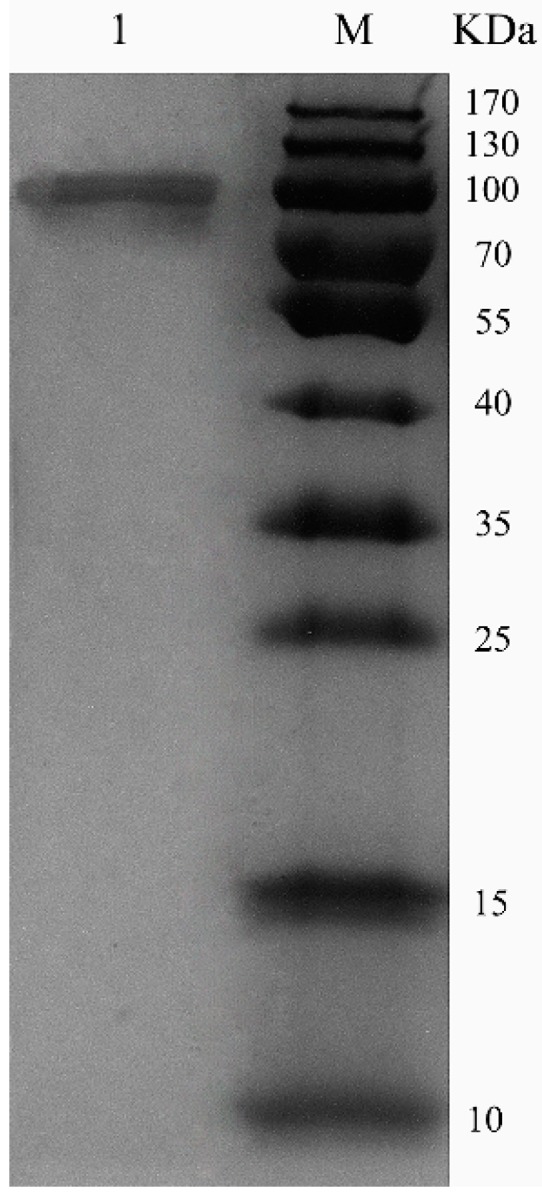
SDS-PAGE analysis of purified ChSase AC II in *P. pastoris* X33. Lane M, 170 kDa protein marker; lane 1, purified ChSase AC II.

**Figure 4 marinedrugs-17-00185-f004:**
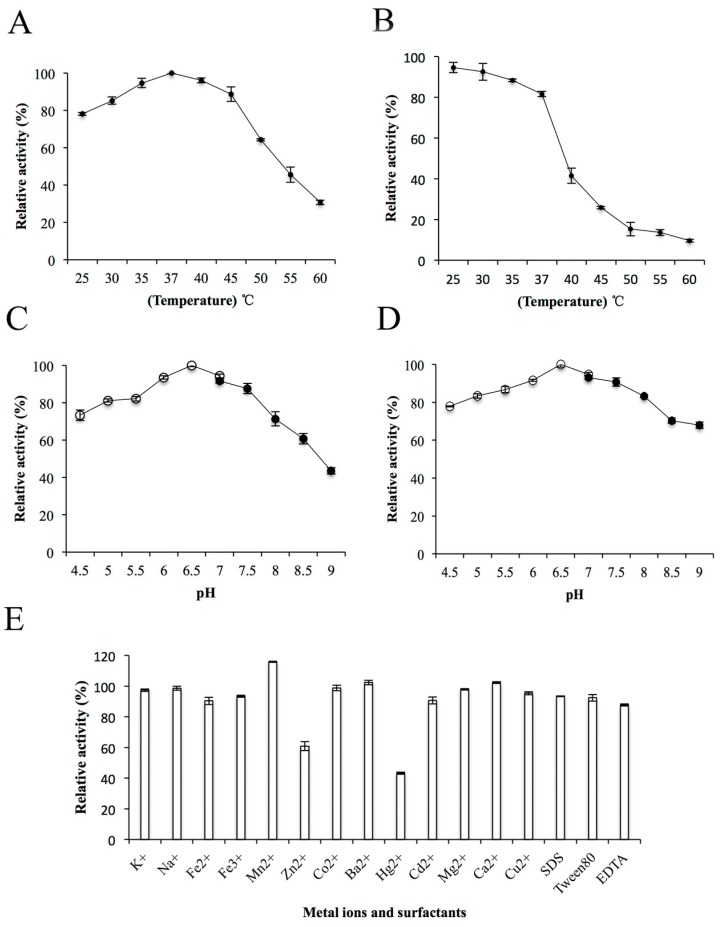
Biochemical characterization of ChSase AC II. (**A**) The optimal temperature of ChSase AC II. (**B**) The thermal stability of ChSase AC II. (**C**) The optimal pH of ChSase AC II. (**D**) The pH stability of ChSase AC II. (**E**) The effect of metal ions and surfactants on ChSase AC II. The highest activity was set at 100%. Each value represents the mean and standard deviation of three replicates.

**Figure 5 marinedrugs-17-00185-f005:**
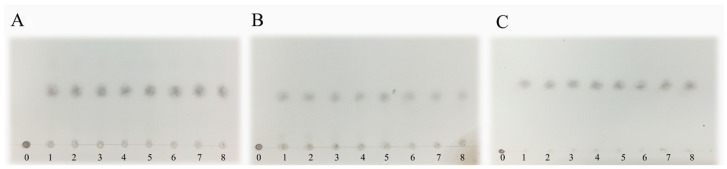
TLC analysis of the degradation products of ChSase AC II towards CS-A, CS-C and HA. Line 0–8: hydrolysates of CS-A (**A**), CS-C (**B**), and HA(**C**) for 0, 5, 15, 30 min, 1, 2, 3, 4, 5 h, respectively.

**Figure 6 marinedrugs-17-00185-f006:**
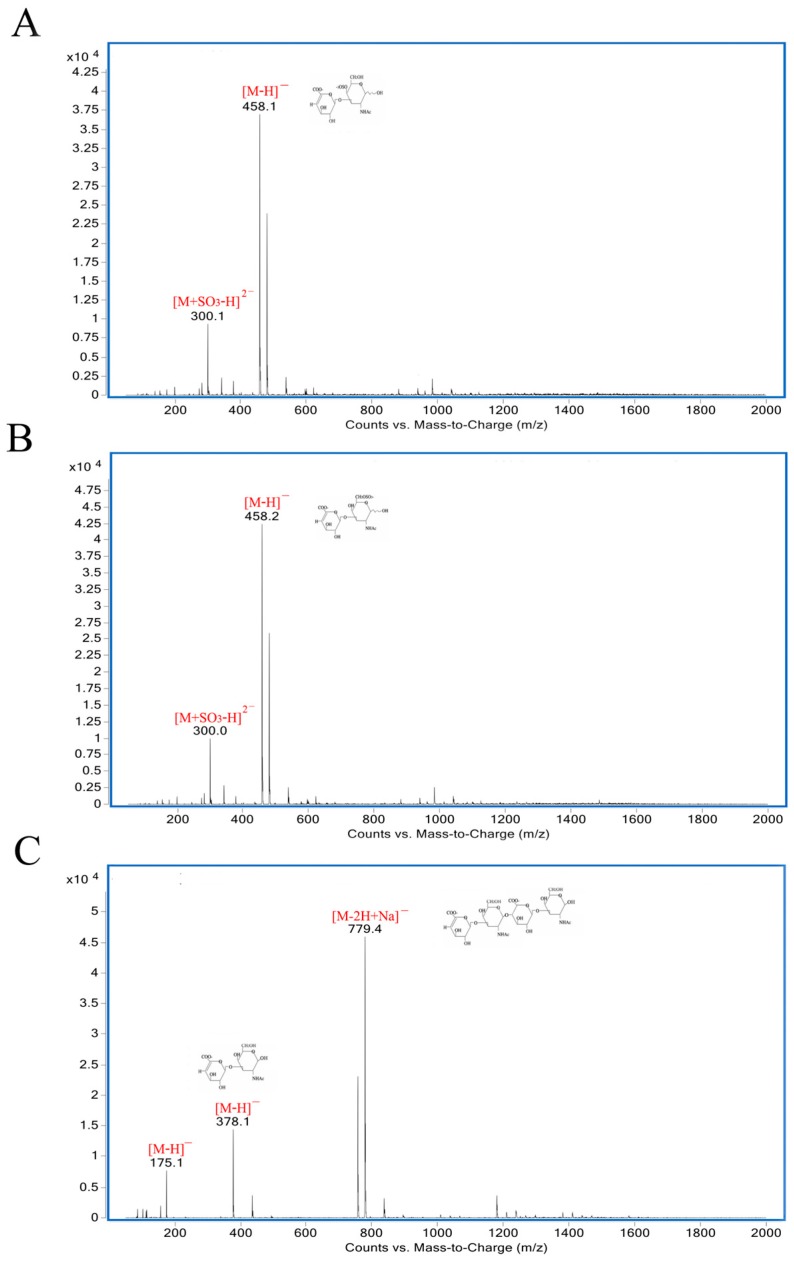
ESI-MS analysis of ChSase AC II hydrolysis products with CS-A (**A**), CS-C (**B**), and HA (**C**) as substrates.

**Figure 7 marinedrugs-17-00185-f007:**
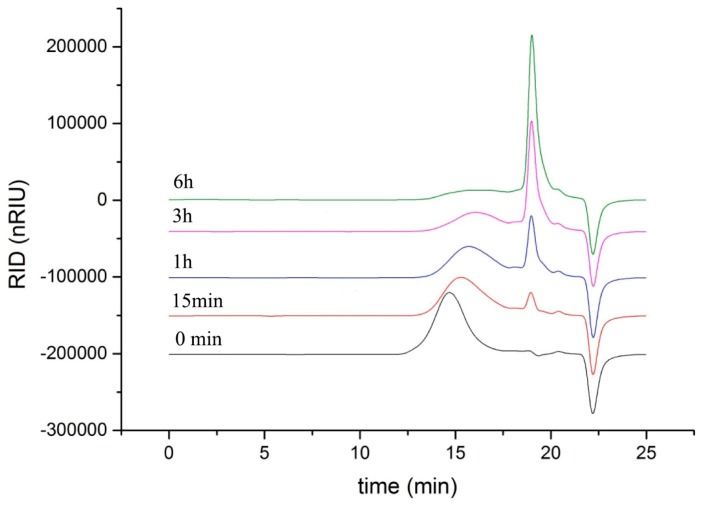
HPLC analysis of the CS-C degradation products of ChSase AC II.

**Table 1 marinedrugs-17-00185-t001:** Summary of the purified ChSase AC II.

Step	Total	Total	Specific Activity	Fold	Yield
	Protein (mg)	Activity (U)	(U/mg Protein)	Purification	(%)
Fermentation media	889.9	2723	3.06	1.0	100.0
Ultrafiltration	480.2	1969	4.10	1.34	72.3
HiTrap QFF column	77.8	1454	18.7	6.11	53.4

**Table 2 marinedrugs-17-00185-t002:** Substrate specificity and kinetic parameters of ChSase AC II.

Substrate	Relative Activity	V_max_	K_m_	K_cat_	K_cat_/K_m_
	(%)	(µM/min)	(µM)	(min^−1^)	(min^−1^/µM)
CS-A	115.64	186	2.44	232	95.1
CS-C	100	168	1.68	210	125
HA	294.93	472	0.165	590	3575

**Table 3 marinedrugs-17-00185-t003:** Media for strains.

Media	Composition
Selective	0.5% CS, 0.5% yeast extract (YE), 0.5% NaCl, 0.2% MgSO4, 0.05% KH_2_PO_4_, pH 7.0
Luria-Bertani (LB)	0.5% YE, 1% NaCl, 1% peptone, pH 7.0
Low Salt LB (LLB)	0.5% YE, 0.5% NaCl, 1% peptone, pH 7.0
Yeast extract peptone dextrose medium (YPD)	1% YE, 2% peptone, 2% glucose, pH 7.0
YPD+sorbitol (YPDS)	1% YE, 2% peptone, 2% glucose, 1 M sorbitol, pH 7.0
Minimal dextrase medium (MD)	2% glucose, 1.34% yeast nitrogen base without amino acids and ammonium sulfate (YNB), 4 × 10^−5^% biotin, pH 7.0
Buffered glycerol-complex medium (BMGY)	1% Glycerol, 1% YE, 2% peptone, 0.34% YNB, 1% (NH_4_)_2_SO_4_ (w/v) and 4 × 10^−5^% biotin in pH 6.0 100 mM potassium phosphate buffer

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
