# Peer review of "Expression, Purification and Characterization of Chondroitinase AC II from Marine Bacterium Arthrobacter sp. CS01"

_marinedrugs, 2019, doi:10.3390/md17030185_

Round 1

Reviewer 1 Report

The authors submitted very interesting manuscript, which deals with enzymatic activities of chondroitinase  AC II. The enzyme was isolated form a marine bacterial genus Arthrobacter. The topic is timely because this enzyme can be used in pharmaceutical industry for oligosaccharide production. These oligosaccharides are often employed for an effective inhibition of degenerative processes associated with osteoarthrosis. 

The authors isolated and characterized the enzyme. They explored the properties of the enzyme with care. The adequate enzymological methods were employed for the enzyme characterisation. The experimental data supports the results, which are properly discussed. Furthermore, the authors cited the most important refences. 

Since I found no serious mistakes in the manuscript, I recommend the paper for publication.

Author Response

Reviewer #1: The authors submitted very interesting manuscript, which deals with enzymatic activities of chondroitinase  AC II. The enzyme was isolated form a marine bacterial genus Arthrobacter. The topic is timely because this enzyme can be used in pharmaceutical industry for oligosaccharide production. These oligosaccharides are often employed for an effective inhibition of degenerative processes associated with osteoarthrosis. The authors isolated and characterized the enzyme. They explored the properties of the enzyme with care. The adequate enzymological methods were employed for the enzyme characterisation. The experimental data supports the results, which are properly discussed. Furthermore, the authors cited the most important refences. Since I found no serious mistakes in the manuscript, I recommend the paper for publication Response : We sincerely thank the kind reviewer very much for your great efforts on our manuscript. We are encouraged to improve further study and paper preparation by your approval.

Reviewer 2 Report

Thsi manuscript reports a chondroitinase isolated fthe marine bacterium Arthrobacter, expressed in  Pichia pastoris and characterized for the mode of action by TLC and ESI-MS analysis. Although of interest for this journal I will suggest rejection for the preparation of a new resubmission where a better manuscript is prepared. English and sentence construction are spotted in different parts of Introduction which make the reading very difficult (line 29.30, 43-45, 54-56, 60-61. Presentation of paragraph 2.6. Analysis of reaction mode and products can be greatly improved by presenting the structures of identified peaks and reaction scheme. Check english at line 130 too. For the presentation of 2.5. Biochemical properties of recombinant ChSase AC II, authors compared different enzymes from other organisms; please check this with the sentence at line 54-56 and give a better explanation. I would also suggest a different location of this manuscript within the long list of special issues of Marine Drugs. Selection of a number of special issue with a different degree of suitability is possible.

Author Response

1. English and sentence construction are spotted in different parts of Introduction which make the reading very difficult (line 29.30, 43-45, 54-56, 60-61). Response: Thank the reviewer for your valuable suggestions. We have revised the spelling and grammatical errors of the manuscript in accordance with the reviewer’s suggestions. In addition, we have asked international students and specialized agency (Editage.cn, Cactus Communications Inc.) for manuscript modification. We are confident now that the language of this manuscript can meet the journal’s requirement. 2. Presentation of paragraph 2.6. Analysis of reaction mode and products can be greatly improved by presenting the structures of identified peaks and reaction scheme. Response: Thanks for the reviewer’s advice and we supplemented the structures of identified peaks in Figure 6. 3. Check english at line 130 too Response: The manuscript has been revised in accordance with the reviewer’s suggestions. 4. For the presentation of 2.5. Biochemical properties of recombinant ChSase AC II, authors compared different enzymes from other organisms; please check this with the sentence at line 54-56 and give a better explanation. Response: Thanks for the reviewer’s suggestion. sentence at line 54-56 show the catalytic mechanism of ChSase according to protein crystallographic analysis, the related reports are little. Since the protein structure and catalytic dynamics are not focused in this study, therefore, biochemical properties of recombinant ChSase AC II, authors compared different enzymes from the reported other organisms. In addition, the research on protein structure and catalytic dynamics of recombinant ChSase AC II are in progress, these result and discussion will be compared the sentence at line 54-56.

Reviewer 3 Report

The authors of the presented manuscript characterized one chondroitin lyase from a marine bacterium that was isolated from sea cucumber gut. Everything important (eg. pH- and temperature profiles, influence of different metal ions, kientic parameters, relatedness to other enzymes of the same kind) was presented. Beyond that even data for different expression hosts is presented. In general, the data seems reasonable and the research was carried out thoroughly. However, I would like to encourage the authors to consider the following main comments to improve their manuscript:

I am lacking a bit more info on the enzymes itself in the introduction. They are introduced briefly  but I would like to read also a bit about the mechanism, which also makes them lyases. This is somewhat important, because you are using the fact of them being lyases in your assays be measuring the absorption of the double bond in the product.

Please prepare a proper purification table (for advice refer to Burgess, R.R. Chapter 4 Preparing a Purification Summary Table. In Methods in Enzymology - Guide to Protein Purification; Burgess, R.R., Deutscher, M.P., Eds.; Elsevier, 2009; Vol. 463, pp. 29–34.). Your table 1 is unfortuantely not very useful.

Please give also kcat/Km in your kinetic parameters. This value is much more comparable.

I am lacking a reaction time course with quantified products. Figure 5 shows product formation only qualitatively and it looks to me like there is nothing happening anymore after 5 min. So I strongly recommend to use the LC-MS you have used to identify your product also to quantify the amounts of product formed over the reaction time.

I wondered about that you used Tris-HCl buffer over the whole pH range from 4.5 - 9 (p.8, l. 197). Tris buffer has only a usefull buffer capacity from pH 7.2 - 9.2. I suggest to use a broad range buffer system (eg. DAVIES buffer) especially for determining pH profiles.

Minor comments:

p.1, l.41: how do you define LMWCS? range of DP?

p.2, l.70 : Please add LHYM225 to the Arthobacter sp.

p.6, l.146: Please add the source of the sea cucumbers

p.7, l.152: How do you isolate the sea cucumber gut? Please specify selective agar plates ingredients (also in line 154)

p.7, l.174: Please specify G418.

p.7, l.188: please specify the unit for the absorption coefficient.

p.8, l.202-203: please specify the counter ions of the metal salts used.

p.8, l.210: Please specify the conditions of the ESI-MS. To me it is not clear if you actually used the LC as well or not.

Author Response

They are introduced briefly  but I would like to read also a bit about the mechanism, which also makes them lyases. This is somewhat important, because you are using the fact of them being lyases in your assays be measuring the absorption of the double bond in the product.

Response: Thanks for the reviewer’s suggestion and we added information about the lyase mechanism. In contrast to the vertebrates, chondroitinase from microorganisms utilize a β- eliminative mechanism to breakdown CS, through splitting β-1,4-glycosidic bonds between β-D-glucuronic acid (GlcA) and N-acetyl-β-D- galactosamine (GalNAc) with concomitant formation of an unsaturated C4-C5 bond within the GlcA located at the nonreducing end[1].

[1] Linhardt, R.J.; Avci, F.Y.; Toida, T.; Kim, Y.S.; Cygler, M. Cs lyases: structure, activity, and applications in analysis and the treatment of diseases. Adv. Pharmacol. 2006, 53, 187-215.

Please prepare a proper purification table (for advice refer to Burgess, R.R. Chapter 4 Preparing a Purification Summary Table. In Methods in Enzymology - Guide to Protein Purification; Burgess, R.R., Deutscher, M.P., Eds.; Elsevier, 2009; Vol. 463, pp. 29–34.). Your table 1 is unfortuantely not very useful.

Response: We agreed with the reviewer’s opinion and made a revision based on the reviewer’s suggestion.

Please give also kcat/Km in your kinetic parameters. This value is much more comparable.

Response: We agreed with the reviewer’s opinion and supplemented relevant data in Table 2.

 Figure 5 shows product formation only qualitatively and it looks to me like there is nothing happening anymore after 5 min. So I strongly recommend to use the LC-MS you have used to identify your product also to quantify the amounts of product formed over the reaction time.

Response: Thanks for the reviewer’s suggestion.

To determine the changes in product formation and substrate consumption during degradation, both the samples were analyzed by HPLC using PL aquagel-OH 30 column (Agilent, USA) under the following conditions: the mobile phase was buffer concluded of NaNO3 (0.2M) and NaH2PO4 (0.01M), the column temperature was 25 °C, the flow velocity was 0.5 mL/min, and the detector was a refractive index detector (RID).

As is shown in figure 7, during CS-C degradation by ChSase AC II, only one peak with a retention time of 18.990 min was increased with reaction time, and the substrate peak(retention time of 15.709 min) was decreased.

I wondered about that you used Tris-HCl buffer over the whole pH range from 4.5 - 9 (p.8, l. 197). Tris buffer has only a usefull buffer capacity from pH 7.2 - 9.2. I suggest to use a broad range buffer system (eg. DAVIES buffer) especially for determining pH profiles.

Response: Thanks for the reviewer’s advice and we are sorry for this inaccurate expression. The pH profiles were determined to be from 4.5 to 7.0 in 20 mM Na2HPO4-citric acid buffer and from 7.0 to 9.0 in 20 mM Tris-HCl buffer.

 p.1, l.41: how do you define LMWCS? range of DP?

Response: Thanks for the reviewer’s question and we usually define chondroitin sulfate below 20kDa as LMCWS, but unfortunately, we have not found a specific literature to specify the range of DP.

  p.2, l.70 : Please add LHYM225 to the Arthobacter sp.

Response: The manuscript has been revised in related sections.

  p.6, l.146: Please add the source of the sea cucumbers.

Response: The sea cucumber was from Weihai sea cucumber farm (Shandong, China).

 p.7, l.152: How do you isolate the sea cucumber gut? Please specify selective agar plates ingredients (also in line 154).

Response: The sea cucumber gut was isolated by Weihai sea cucumber farm (Shandong, China). Then the sea cucumber were immersed, homogenated, and diluted with sterilized sea water and spread on selective agar plates in our laboratory. And the selective agar plates ingredients were in Table 3 from supporting information.

 p.7, l.174: Please specify G418.

Response: G418 is Geneticin, used as selective antibiotic. The manuscript has been revised in related sections.

 p.7, l.188: please specify the unit for the absorption coefficient.

Response: Thanks for the reviewer’s question. a millimolar absorption coefficient of 5.1 L/(mol·cm)was used in the calculations. The manuscript has been revised in related sections.

  p.8, l.202-203: please specify the counter ions of the metal salts used.

Response: Thanks for the reviewer’s question and we used following reagents: 10 mM KCl, NaCl, FeCl2, FeCl3, MnCl2, ZnCl2, CoCl2, BaCl2, HgCl2, CdCl2, MgCl2, CaCl2 and CuCl2.

p.8, l.210: Please specify the conditions of the ESI-MS. To me it is not clear if you actually used the LC as well or not.

Response: we used negative-ion ESI-MS (Agilent 1290I nfinity II-6460, Frag=175.0V, m/z 100-2000 amu) and added the LC result of degradation products in the manuscript.

Round 2

Reviewer 2 Report

Quality of presentation of the interesting topic of this manuscript has been greatly improved. One last remark is at the line 230 of the new version where the authors report the details of mass spectroscopic studie, they used the word "both" without any significance in my opinion. They have ten samples to be analyzed not only two and if the meaning of "both" intended was reaction sample and blank sample this can be explicated in a better manner. After this little modification the manuscript can be accepted.

Author Response

Point 1. Quality of presentation of the interesting topic of this manuscript has been greatly improved. One last remark is at the line 230 of the new version where the authors report the details of mass spectroscopic studie, they used the word "both" without any significance in my opinion. They have ten samples to be analyzed not only two and if the meaning of "both" intended was reaction sample and blank sample this can be explicated in a better manner. After this little modification the manuscript can be accepted.

Response 1: Thank the reviewer for your opinion through rigorous thinking, The sentence has been revised as following : “To determine the changes in product formation and substrate consumption during degradation, samples taken at different reaction time(0 min, 15 min, 1 h, 3 h, 6 h) were analyzed by HPLC......”